# Surrogate Modelling for Oxygen Uptake Prediction Using LSTM Neural Network

**DOI:** 10.3390/s23042249

**Published:** 2023-02-16

**Authors:** Pavel Davidson, Huy Trinh, Sakari Vekki, Philipp Müller

**Affiliations:** 1Faculty of Information Technology and Communication Sciences, Tampere University, 33720 Tampere, Finland; 2Faculty of Sport and Health Sciences, University of Jyväskylä, Seminaarinkatu 15, 40014 Jyväskylän yliopisto, Finland

**Keywords:** oxygen uptake, INS/GPS, running metrics, machine learning, LSTM neural network

## Abstract

Oxygen uptake ($\dot{V}O$${}_{2}$) is an important metric in any exercise test including walking and running. It can be measured using portable spirometers or metabolic analyzers. Those devices are, however, not suitable for constant use by consumers due to their costs, difficulty of operation and their intervening in the physical integrity of their users. Therefore, it is important to develop approaches for the indirect estimation of $\dot{V}O$${}_{2}$-based measurements of motion parameters, heart rate data and application-specific measurements from consumer-grade sensors. Typically, these approaches are based on linear regression models or neural networks. This study investigates how motion data contribute to $\dot{V}O$${}_{2}$ estimation accuracy during unconstrained running and walking. The results suggest that a long short term memory (LSTM) neural network can predict oxygen consumption with an accuracy of 2.49 mL/min/kg (95% limits of agreement) based only on speed, speed change, cadence and vertical oscillation measurements from an inertial navigation system combined with a Global Positioning System (INS/GPS) device developed by our group, worn on the torso. Combining motion data and heart rate data can significantly improve the $\dot{V}O$${}_{2}$ estimation resulting in approximately 1.7–1.9 times smaller prediction errors than using only motion or heart rate data.

## 1. Introduction

Oxygen consumption ($\dot{V}O$${}_{2}$) during unconstrained running can be measured directly by portable spirometers or metabolic analyzers. However, using these devices on a constant basis is not convenient and often requires trained personnel. Furthermore, the price range of portable spirometers can be in tens of thousands of euros, causing additional operational costs. Therefore, estimating $\dot{V}O$${}_{2}$ based on measurements from consumer-grade sensors and without reliance on direct measurements would be highly useful for different performance assessment applications and enable their wide-range use.

Indirect estimation of $\dot{V}O$${}_{2}$ from the observation of other variables (using surrogate variables) is a growing field of research thanks to the development of small wearable sensors and machine learning algorithms. Typically, surrogate modeling of instantaneous oxygen uptake or its steady value is based on heart rate (HR) measurements [1]. However, there are factors that can affect the relationship between heart rate and oxygen uptake. Some of these include hydration status, exercise duration, medication, altitude, state of training, environmental conditions, and time of day [2]. Therefore, some additional input features can be considered, and their choice depends on specific application. For example, in running it could be breathing frequency, speed, speed variation and cadence that can be computed by wearable devices during outdoor exercise [3]. For cycling, in addition to HR and breathing frequency, the input features can include mechanical power and pedaling cadence that can be accurately measured directly on cycling ergometers [4,5,6].

Many previous studies used linear regressions for $\dot{V}O$${}_{2}$ prediction [1,3,7,8]. However, this approach has limitations for very low and very high intensity exercises when the HR vs. $\dot{V}O$${}_{2}$ relationship is significantly non-linear. Furthermore, heart rate is affected by factors such as day-to-day variability, age, sex, fitness, exercise modality, and environmental conditions [3]. Due to a reduced number of parameters, analytical models cannot adapt to changes in exercise conditions. To mitigate the effect of these variables, HR index (HRI) can be used as a surrogate for $\dot{V}O$${}_{2}$ prediction instead of HR [1,8]. HRI is equal to a given HR divided by resting HR and it is considered that it can potentially remove the need for individual calibration often required for tracking daily activity using HR [8].

Some commercial products, for example, the Suunto personal HR monitoring system, use HR measurements as a surrogate for the estimation of $\dot{V}O$${}_{2}$ and energy expenditure [7]. In this approach the input features include R-wave-to-R-wave (R-R) heartbeat intervals, R-R–derived respiration rate, and the on-and-off $\dot{V}O$${}_{2}$ dynamics during various exercise conditions. Although the investigators acknowledge the limitations in the prediction accuracy when individual values for maximal HR and $\dot{V}O$${}_{2}$ are included, they give little information on the validity against pulmonary gas exchange values or correction factors to account for variation in these estimates.

In [9], machine learning was used to predict $\dot{V}O$${}_{2}$ during walking and with other daily activities using surrogate variables such as HR, breathing frequency, minute ventilation, hip acceleration and walking cadence. ECG and respiration band were integrated into a Hexoskin smart shirt. Walking cadence was computed based on hip acceleration using proprietary algorithm. From the HR data, a new variable was derived. The ΔHR was composed by the difference between the current HR value and the previous value by a 1 s lag operator, capturing dynamic changes in cardiac activity. The $\dot{V}O$${}_{2}$ data were measured breath-by-breath by a portable metabolic system (K4b2, COSMED, Italy). The predicted oxygen uptake $\dot{V}O$${}_{2}$ was obtained by a random forest regression based on these features streamed from wearable sensors throughout the day. The deployment of such nonintrusive technologies can help us to study relationships between patterns of daily physical activity and fitness markers. The prediction accuracy was 6.166 mL/min/kg (95% limits of agreement).

For $\dot{V}O$${}_{2}$ prediction, advanced machine learning techniques can be used, such as long-short term memory (LSTM) neural networks (e.g., in [4], a recurrent neural network with three LSTM layers is used). These networks can process entire sequences of data and consider the historical context of inputs. This property is important for applications such as oxygen uptake predictions. Therefore, LSTM networks are widely used in the field of exercise physiology to model $\dot{V}O$${}_{2}$ dynamics across a variety of exercise conditions. They perform better than analytical models or linear regressions [4]. In [4], an LSTM neural network was trained from laboratory cycling data collected on electromagnetically-braked bicycle ergometer (Excalibur Sport, Lode) to predict $\dot{V}O$${}_{2}$ values from easy-to-obtain inputs, such as heart rate, mechanical power output, cadence, and respiratory frequency. Accuracy of workload was 2% for power between 100 and 1500 Watt. The authors compared the performance of their LSTM neural network to that of two analytical models: (a) a first order model assuming $\dot{V}O$${}_{2}$ dynamics to be linear and (b) the model from [10]. The root mean squared error (RMSE) of the LSTM was about 7.2% of the $\dot{V}O$${}_{2max}$. For the analytical models the respective RMSEs were between 10 and 15% of the $\dot{V}O$${}_{2max}$, depending on the test intensity. One drawback of neural networks is that model parameters often do not have direct physiological meaning and require big datasets for training.

Amelard et al. [5] investigated the temporal prediction of $\dot{V}O$${}_{2}$ from wearable sensors during cycle ergometer exercise using a temporal convolutional neural network (TCN) or stacked LSTM. The four cardiorespiratory bio-signals (HR, HR reserve, breathing frequency, and minute ventilation) derived from a Hexoskin smart shirt (Hexoskin, Carre Technologies, Montreal, Canada). The shirt contained a textile electrocardiogram to measure HR and thoracic and abdominal respiration bands to obtain estimates of breathing frequency and minute ventilation (V${}_{E}$) via respiratory inductance plethysmography. The V${}_{E}$ was also calibrated by linear regression to the known V${}_{E}$ measured throughout each protocol with the bi-directional turbine [5]. In addition to these four inputs, the work rate profiles were used as inputs into neural network. The $\dot{V}O$${}_{2}$ data were measured breath-by-breath by a portable metabolic system (MetaMax 3B-R2, CORTEX Biophysik, Leipzig, Germany). TCN showed a $\dot{V}O$${}_{2}$ prediction accuracy of approximately 8% of the $\dot{V}O$${}_{2max}$ (95% limits of agreement) when the input data history was 218 s.

In this study, we investigated how motion data can contribute to $\dot{V}O$${}_{2}$ prediction and if it can be sufficient for accurate $\dot{V}O$${}_{2}$ prediction. We also check whether the addition of heart rate as an input feature can improve $\dot{V}O$${}_{2}$ prediction accuracy. We have developed a data-driven surrogate modelling for accurate oxygen uptake prediction based on motion data measured by an inertial navigation system (INS) combined with a Global Positioning System (GPS) receiver [3] and/or heart rate using an LSTM network. The $\dot{V}O$${}_{2}$ predictions are individualized, i.e., training and predictions are for the same person.

Our measurement setup was based on an INS/GPS device attached to the torso [3] and a heart rate monitor (HRM). We developed a system that can provide continuously different walking and running metrics. We have already used this system for indirect estimation of vertical ground reaction force (vGRF), ground contact time (GCT), and some other target features that can be directly measured only by instrumented insoles or by force plates [3,11]. The results showed that the LSTM network can accurately predict vGRF and GCT based on measurements of accelerations and angular velocities [11].

Considering the promises of LSTM networks, the purpose of this study was to predict the individual response of $\dot{V}O$${}_{2}$ during unconstrained walking and running using measurement from the body-mounted motion sensor and/or HRM. One research goal was to study how motion data contribute to $\dot{V}O$${}_{2}$ prediction and if the motion data alone can achieve a prediction accuracy comparable to HRM based predictions. We also tried to select an optimal LSTM NN model: input features and length of input sequences, number of LSTM layers and hidden units. Agreement between the measured and predicted oxygen uptake is validated by Bland–Altman analysis by computing root mean square errors.

## 2. Materials and Methods

### 2.1. Experimental Data

Twelve recreational runners (age 30.9 ± 8.7 yrs, height 176.9 ± 8.9 cm, body mass 78.3 ± 13.0 kg, body mass index 24.9 ± 3.1 kg/m${}^{2}$, nine males) with varying fitness levels participated in the field tests on a level outdoor track. All participants were healthy and trained, according to their own statements, on average two times per week. The Ethics Committee of Tampere Region approved the study. The participants gave informed consent, and the research was conducted in accordance with the WMA Declaration of Helsinki [12]. After arriving at the track and providing consent, the participants were equipped with the chest mounted INS/GPS device, oxygen mask, gas analyzers and telemetry radios from Oxycon Mobile and, in some cases, a heart rate monitor’s chest strap. The setup is shown in Figure 1. Participants were instructed to walk or run on the outdoor track at speeds ranging between 1.3 and 3.5 m/s. The speed was chosen subjectively based on the participant’s own feeling of fast and slow pace.

Every test for each test person started from resting condition (sitting or standing still) for 3 min to measure oxygen consumption at rest. This allowed us to see the effect of exercise on oxygen consumption. The test continued with walking at two different speeds and running at two different speeds. Thus, data from four speeds and two different gaits were collected. The participants maintained each speed for 3 min so that the oxygen consumption had time to reach a steady state. In between each tested speed, the participants were standing still to allow oxygen uptake and heart rate to return to resting level. The total duration of tests for one participant was about 20 min. The data were collected in two measurement campaigns, on 24/25 May and 1/2 November 2022 in Jyväskylä, Finland. The field tests were carried out at different temperatures, ranging from +3 ${}^{{}^{\circ}}$C up to +22 ${}^{{}^{\circ}}$C, and different relative humidity levels, ranging from 21% to 100%.

Temperature has significant impact on oxygen consumption. For example, in [13] authors tested with eleven healthy subjects at various temperatures. They found that, on average, oxygen consumption was 20.5% higher at −10 ${}^{{}^{\circ}}$C compared to 20 ${}^{{}^{\circ}}$C at same exercise intensities. Similarly, Sandsund et al. [14] discovered that at −15 ${}^{{}^{\circ}}$C oxygen consumption was 10.8% higher than at 23 ${}^{{}^{\circ}}$C during submaximal exercise intensities. They tested with eight elite non-asthmatic cross-country skiers. Regarding the above mentioned percentages it is crucial to note that the impact of temperature decreases as exercise intensity increases due to the body’s increased heat production [15]. Oksa et al. [16] argue that the increase in oxygen consumption at lower temperatures may be caused by “shivering or increase thermoregulatory tonus of the muscles”, altered neuromuscular function and use of additional clothing, which increases the metabolic rate.

Influence of humidity on oxygen uptake was investigated in [17]. The authors of this study found no difference in $\dot{V}O$${}_{2}$ at four sub-maximal velocities of 2.7, 3.3, 3.9, and 4.5 m/s, respectively, for 4 min per stage during tests with different levels of relative humidity (23, 43, 52, 61 and 71%) at 31 ${}^{{}^{\circ}}$C temperature. As our measurements were taken in May at 20 ${}^{{}^{\circ}}$C/40% relative humidity (https://www.jyv-weather.info/wxhistory.php?date=202205, (accessed on 13 January 2023)) and in November at 5 ${}^{{}^{\circ}}$C/90% relative humidity, the absolute humidity, i.e., the total mass of water vapor present in the air were almost the same. Thus, in our measurements humidity has no considerable impact on the $\dot{V}O$${}_{2}$ measurements.

$\dot{V}O$${}_{2}$ measurements and respiratory frequency were collected using breath-by-breath methods from a portable spirometer (Oxycon Mobile, Jaeger, Würzburg, Germany) every 5 s. Immediately before every test session, the gas analyzer and the flow meter were calibrated. The data from the Oxycon Mobile were collected on the notebook using provided telemetry. The accuracy of Oxycon Mobile was discussed in [18]. According to the authors of this study, the Oxycon Mobile significantly underestimates $\dot{V}O$${}_{2}$ at high workloads above 200 W. Typical bias is about 100 mL/min and standard deviation is approximately 125 mL/min. The oxygen uptake measurements were post-processed to reduce the measurement noise and remove outliers. A Savitzky-Golay filter [19] with polynomial order and frame length set to 3 and 1, respectively, was applied three times on the oxygen input data (Figure 2). HR measurements were recorded continuously (beat-by-beat) during the test with a Suunto Movesense IMU+HR system with an output rate of approximately 1 Hz. It was also smoothed and interpolated after the tests.

Measurements of running and walking parameters were collected continuously by the INS/GPS datalogger and saved to a memory card through a wired connection, which ensured that no data were lost. For our outdoor walking and running tests on a level track, the datalogger unit was attached to the torso of test subjects. After the experiment, the data were transmitted to cloud storage using a 4G/LTE USB modem connected to the datalogger. A detailed description of this setup is provided in [3]. The accuracy of speed and speed difference is approximately 0.05 m/s. Vertical displacement and step duration are computed with the accuracy of about 0.01 m and 10 ms, respectively. The output frequency for acceleration, velocity, angular velocity and orientation is 400 Hz. To compute walking and running metrics, the step segmentation was performed. The metrics were computed with the step frequency for each step.

### 2.2. Dataset Preparation

Oxygen uptake, heart rate and motion measurements were synchronized in time in a post-processing phase. Jumps with both feet were executed at the beginning of each data recording simultaneously with bookmarks in oxygen uptake measurements, to obtain time-synchronization between the three different devices. The jumps caused sharp peaks in the acceleration measured both by the INS/GPS and the Suunto Movesense IMU+HR systems. Using these peaks, the synchronization of oxygen uptake, heart rate and motion measurements, as well as the data analysis, were performed offline on a computer after the experiment. The achieved accuracy in time synchronization of about 0.1–0.2 s is sufficient because HR and $\dot{V}O$${}_{2}$ are slowly changing variables with sampling rates of 1 Hz and 0.2 Hz, respectively.

The motion data were processed to obtain step segmentation and compute running and walking metrics for each step as described in [3]. Accelerations and velocities were computed in the anatomical frame. Once the data were segmented into steps, features or metrics that are commonly used in walking and running were computed for each step to facilitate analysis of the data (see [3,11] for details). The oxygen uptake and HR measurements were resampled to meet the same step-by-step frequency as the running metrics. To use an LSTM neural network for oxygen uptake prediction, a training set of sequences (input features) and target values (oxygen uptake) must be created. The following running metrics were computed for each step and selected as input features in this study:Speed averaged over one step: arithmetic mean of the speed (=step length/duration of step), m/s;Speed: peak-to-peak difference during one step, m/s;Step duration, s;Vertical displacement: peak-to-peak difference in vertical movement, m;Heart rate, bpm.

The number of steps in input sequences is one of the hyperparameters that relates the length of past surrogate variables and oxygen uptake. Based on experiments, we decided that input sequences of 50 steps yield a good estimate of the time-dependence decay between the output and past values of inputs. For a better fit, and to prevent the training from diverging, the input sequences were normalized between 0 and 1.

In machine learning, optimal feature selection is crucial for developing simple yet reliable prediction models. The purpose of feature engineering is to identify valid, useful, and understandable patterns in INS/GPS data that have a strong correlation with the target parameter ($\dot{V}O$${}_{2}$), but minimum inter-correlation with the other features. Figure 3 shows the absolute correlations between pairs of input features and target features averaged for all participants.

These recommendations are not binding. Sometimes features that are weakly correlated with the target features can significantly improve the machine learning algorithm prediction accuracy if they are combined with other features. Furthermore, one needs to keep in mind that weak correlation between the input and the target features indicates only that there is no linear relationship between the two features but provides no information about potential non-linear relations.

The selected set of input features was validated based on the consider only one and leave-one-out approaches. It shows that speed, speed change and heart rate are the most important input vectors to predict the $\dot{V}O$${}_{2}$ label. Adding vertical displacement and step duration improves the predictions to some extent. Adding further input features does not improve the prediction accuracies. Figure 3 shows that speed and oxygen uptake are correlated. However, if speed or HR is the only input feature the accuracy is worse than when they are combined with other features. The optimal set of input features was selected based on correlation matrix and our domain knowledge. The entire dataset was shuffled and split into training and test subsets with 80% to 20% ratio.

### 2.3. LSTM Network Architecture

Oxygen uptake predictions have to take into account the historical context of inputs, therefore a many-to-one long-short term memory (LSTM) model was developed for oxygen uptake ($\dot{V}O$${}_{2}$) prediction. Optimal length of the input sequences is estimated based on the results. We tried sequences of 50 and 100 steps and concluded that shorter sequences are better. The neural network was implemented in Matlab Deep Learning toolbox. The neural network includes the following layers: one LSTM layer, input, output and dense layers. We used a sequence input layer that can contain two to five input sequences and its input size matches the number of channels of the input data. The LSTM layer has 150 hidden units. The number of hidden units determines how much information is learned by the layer. Larger values can yield more accurate results but can be more susceptible to overfitting to the training data. The output is a single number for each set of sequences giving the value for predicted $\dot{V}O$${}_{2}$. A fully connected layer with a size matching the number of predictors, followed by a regression layer, were used to specify the number of values to predict. The number of LSTM layers is a hyperparameter and was selected based on experiments. The total number of trainable parameters was 93,151 that is far more than training samples (typically 1500–5000). However, LSTM networks work very well despite these potential overfitting problems, mainly because of various regularization effects implicit to the training/optimization algorithm. The obvious benefit of having many parameters is that the network is flexible enough to represent the desired mapping. Adapting the size of the network to the size of the training set can lead to a problem when the network is too simple and unable to represent the desired mapping (high bias).

It was trained using the Adam optimizer for 8000 epochs. For larger datasets, you might not need to train for as many epochs for a good fit. The learning rate was set to 0.005. Training the neural network required approximately 3–5 h on a PC equipped with an Intel Core i7-6700 @3.4 GHz CPU processor. Testing the models requires only a few seconds for every simulation.

To assess the prediction ability of the different models, a residual analysis was conducted. Residuals were calculated as the difference between the experimental $\dot{V}O$${}_{2}$ values and the output $\dot{V}O$${}_{2}$ values predicted by the models. The RMSE of the residuals was calculated. A Bland–Altman analysis was used to assess the level of agreement between measured and predicted data. The mean bias and the limits of agreement at 95% of probability (two times standard deviation) were calculated.

## 3. Results

### 3.1. Input Features Include Only Motion Parameters

Two different input feature options were compared here: (a) four motion parameters including speed, speed change (peak-to-peak) during one step, step duration and vertical oscillation; (b) only speed. The Bland–Altman analysis of the predicted $\dot{V}O$${}_{2}$ using LSTM network with four motion input features and directly measured $\dot{V}O$${}_{2}$ across all exercise conditions and participants combined are shown collectively in Figure 4. Dashed horizontal lines represent 95% limits of agreement and the solid line represents the prediction bias. Each color represents data from a unique participant in the test set. This plot shows that the prediction bias is −0.04 mL/min/kg (approximately 0.2% of $\dot{V}O$${}_{2peak}$) and the validity of the predicted $\dot{V}O$${}_{2}$ values from the LSTM network expressed with 95% limits of agreement is 2.49 mL/min/kg (approximately 5% of $\dot{V}O$${}_{2peak}$).

Example of $\dot{V}O$${}_{2}$ prediction accuracy for a single representative athlete is shown in Figure 5. The plot shows $\dot{V}O$${}_{2}$ measurements (blue) and predictions (red). Based on our tests with 11 participants, it can be concluded that the bias is typically small, less than 0.5 mL/min/kg (<1% of $\dot{V}O$${}_{2peak}$). The standard deviation (σ) is typically about 1.2–1.5 mL/min/kg, 95% of all predictions are within 2σ.

The Bland–Altman analysis of the predicted $\dot{V}O$${}_{2}$ using LSTM network with one input feature (speed) and directly measured $\dot{V}O$${}_{2}$ across all exercise conditions and participants combined are shown collectively in Figure 6. Performance of the same LSTM network for a single representative athlete is shown in Figure 7. The LSTM network with four input features yields more accurate prediction than the LSTM network with only one feature: 2.49 vs. 5.79 mL/min/kg (95% limits of agreement). Although the vertical displacement, speed change and step duration cannot be used alone, they significantly improve the prediction accuracy if they are combined with speed (Figure 5).

### 3.2. Input Features Include Only Heart Rate

When heart rate is the only input feature to the LSTM network, the Bland–Altman analysis of the predicted and directly measured oxygen uptake across all exercise conditions and participants combined is shown collectively in Figure 8. This plot shows that the prediction bias is 0.16 mL/min/kg (approximately 0.35% of $\dot{V}O$${}_{2peak}$) and the validity of the predicted $\dot{V}O$${}_{2}$ values from the LSTM network expressed with 95% limits of agreement is 2.52 mL/min/kg (approximately 5.5% of $\dot{V}O$${}_{2peak}$). Performance of the regressor for a single representative athlete during 20 min-long test is shown in Figure 9.

### 3.3. Input Features Include Motion Parameters and Heart Rate

When the input features to the LSTM network include both the motion features and heart rate, the Bland–Altman analysis of the predicted and directly measured oxygen uptake across all exercise conditions and participants combined is shown collectively in Figure 10. This plot shows that the prediction bias is 0.02 mL/min/kg (approximately 0.05% of $\dot{V}O$${}_{2peak}$) and the validity of the predicted $\dot{V}O$${}_{2}$ values from the LSTM network expressed with 95% limits of agreement is 1.36 mL/min/kg (approximately 3% of $\dot{V}O$${}_{2peak}$). Performance of the regressor for a single representative athlete during 20 min-long test is shown in Figure 11.

## 4. Discussion

Our hypothesis was that an LSTM neural network could be used to accurately predict individual’s oxygen uptake during walking and running from measurements of motion parameters and heart rate. We also wanted to examine how heart rate measurements could improve $\dot{V}O$${}_{2}$ prediction accuracy if they are added to the wearable INS/GPS device that measures acceleration, velocity, angular velocity and orientation of the upper body where the device is attached.

To the best of our knowledge, we are the first to apply recurrent neural networks for prediction of $\dot{V}O$${}_{2}$ during unconstrained walking and running based on measurements of motion parameters only and motion parameters combined with the heart rate. The results show that performance of the LSTM network with the input features that include only motion parameters and the LSTM network with heart rate as the only input are comparable: the bias is −0.04 mL/min/kg vs. −0.16 mL/min/kg and the 95% limits of agreement is 2.49 mL/min/kg vs. 2.52 mL/min/kg. In the first case the bias is negligible. If the motion features are combined with heart rate the LSTM network’s performance is considerably better: the bias is 0.02 mL/min/kg and the 95% limits of agreement is 1.36 mL/min/kg.

All previous studies used heart rate and sometimes mechanical power (in cycling ergometer case) for oxygen uptake prediction. In such approaches, mechanical power is a very important feature. However, it cannot be measured for walking and running directly as it is measured on cycling ergometer. For example, Zignoli et al. [4] studied the ability of recurrent neural networks to predict oxygen uptake during exercises on cycling ergometer. They show that, using heart rate, mechanical power output, pedaling cadence and respiratory frequency as input features to the LSTM network, it is possible to estimate $\dot{V}O$${}_{2}$ with the accuracy of 7.2% of $\dot{V}O$${}_{2max}$ which is approximately 3.5 mL/min/kg. The performance of our approach for $\dot{V}O$${}_{2}$ prediction during unconstrained walking and running is better (1.47 mL/min/kg). Some studies did not use the mechanical power, but the prediction was less accurate. For example, Amelard et al. [5] developed temporal convolutional neural network to predict $\dot{V}O$${}_{2}$ during cycle ergometer exercise using only cardiorespiratory bio-signals (HR, HR reserve, breathing frequency, and minute ventilation). The prediction accuracy was 8% of $\dot{V}O$
${}_{2max}$ which is approximately 3.8 mL/min/kg.

As our LSTM model’s performance was mainly evaluated using data collected during the same session, it remains unclear how well it can perform with data collected during different days. There are many factors that can influence daily oxygen uptake during exercise, such as previous physical activity, state of health and contents and the time of previous meal. Even when these things are taken into account, day-to-day variation in oxygen consumption can be around 5% when performing exercise under a constant load [20]. When thinking about changes in oxygen consumption during exercise in the long run, factors such as endurance training, resistance training and nutrition can also have an effect [21,22].

In our approach, the model training is based on data from a single person and prediction is performed for the same person, similar to other studies [4,5]. Despite of this limitation, our approach still has practical importance. Once the training is complete, the system can be used for several months until a brief update is required. During this time, without using expensive and cumbersome equipment (portable spirometers), it can estimate fitness level, physical conditions, and shape (including potential injuries). To develop a robust model that is stable with respect to physiological variability, the training data have to include measurements from several tests carried out at different temperatures and times over a long period of time. The development of generalizable models for inter-subject (i.e., using trials of some subjects in model training, and taking the trials of different subjects for validation) oxygen uptake prediction is still an open research question.

## 5. Conclusions

Our study suggested that an LSTM network can predict oxygen consumption based only on sequences of motion data collected from wearable sensors with an accuracy of 2.49 mL/min/kg (95% limits of agreement). Motion input features have to include speed and speed change. The addition of vertical oscillations and step duration does not provide a substantial improvement. Physiological markers, such as heart rate, can be added to the input and improve the $\dot{V}O{}_{2}$ prediction accuracy (1.36 mL/min/kg 95% limits of agreement). Achievable accuracy is comparable with the measurement accuracy of an Oxycon Mobile (approximately 250 mL/min 95% limits of agreement) portable spirometer when the training and predictions are for the same person and performed at similar temperatures. Based on this study and our experience with $\dot{V}O{}_{2}$ prediction, we think that this algorithm has the potential to be embedded in a portable system and to provide real-time assessment of individual $\dot{V}O{}_{2}$ during walking and running. The proposed approach for $\dot{V}O{}_{2}$ prediction can provide a unique opportunity for continued $\dot{V}O{}_{2}$ collections in unsupervised environments. Our approach can be applied in continuous assessment of energy expenditure and aerobic fitness with the potential for future applications such as the early detection of possible injuries and the deterioration of physical health. The proposed algorithm can also be adapted to estimate energy expenditure and the quantification of training intensity. We investigated exercising conditions at low and moderate intensities. More work is needed to cover all different conditions including heavy and severe intensity exercises. The effect of inter-subject variation and day-to-day variability on $\dot{V}O{}_{2}$ prediction has yet to be studied.

## Figures and Tables

**Figure 1 sensors-23-02249-f001:**
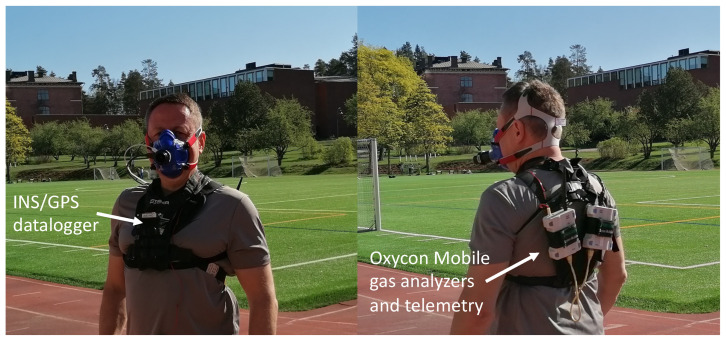
Measurement setup: Oxycon Mobile portable spirometer (the gas analyzers are on the back) and chest mounted INS/GPS device. The heart rate monitor is worn under the shirt.

**Figure 2 sensors-23-02249-f002:**
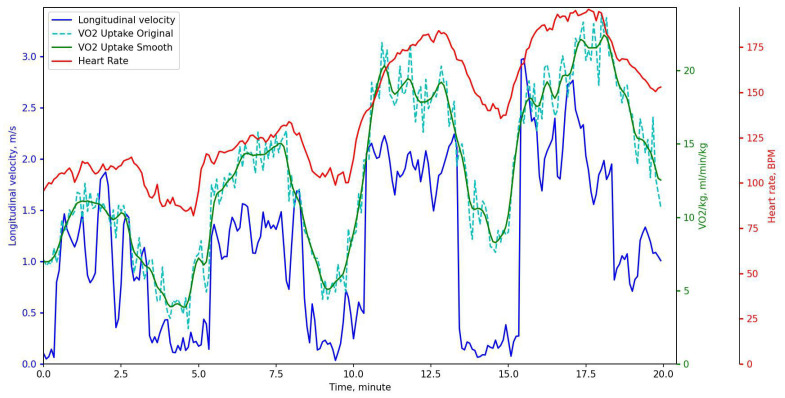
Example of oxygen uptake measurements (dashed green), smoothed oxygen uptake (solid green), HR measurements (red) and speed (blue) during the typical 20 min test.

**Figure 3 sensors-23-02249-f003:**
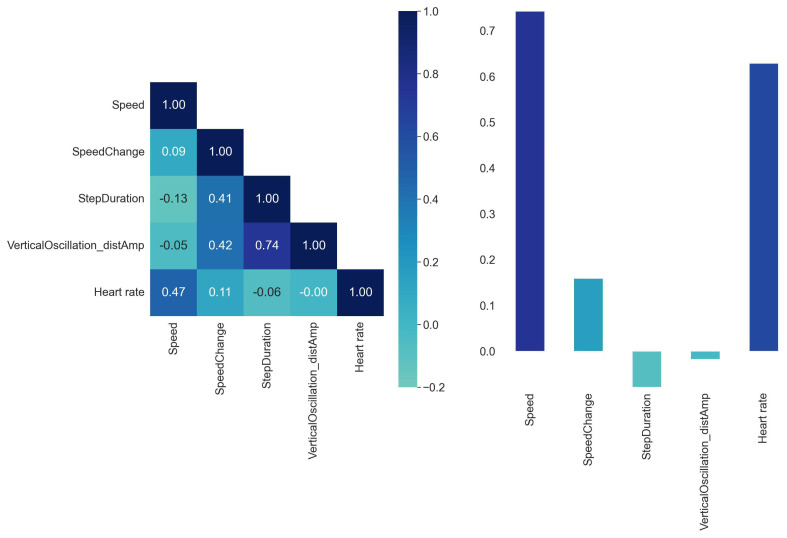
Matrix of correlations between the input features (**left**) and correlations of the input features with the target feature (**right**).

**Figure 4 sensors-23-02249-f004:**
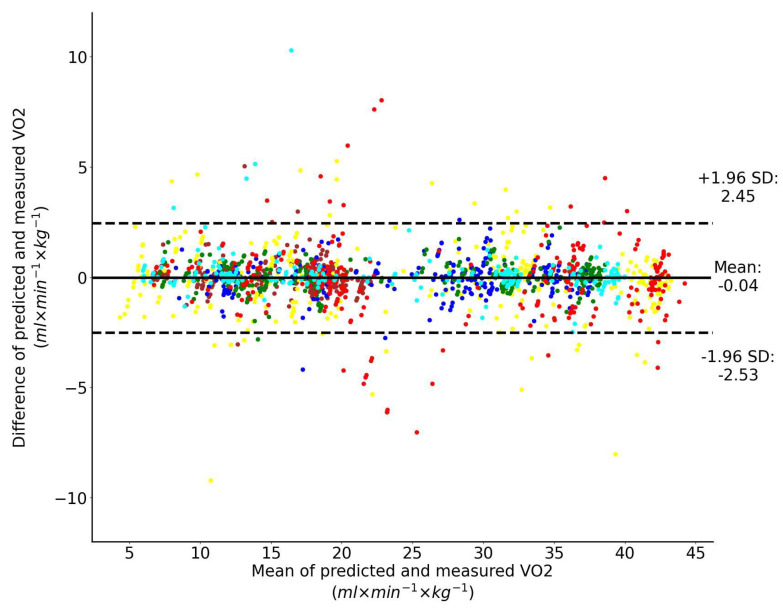
Bland–Altman analysis of the predicted and directly measured oxygen uptake with all exercise conditions combined. The inputs to the LSTM network include four motion features. Dashed horizontal lines represent the 95% limits of agreement and the solid line represents the prediction bias. Each color represents data from a unique participant in the test set.

**Figure 5 sensors-23-02249-f005:**
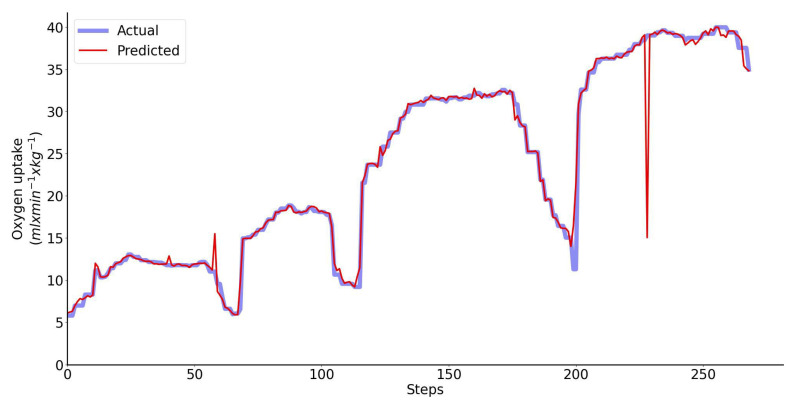
Performance of the regressor is shown for a single representative athlete: predicted $\dot{V}O{}_{2}$ (red) vs. measured $\dot{V}O{}_{2}$ (blue) oxygen uptake. The inputs to the LSTM network include four motion features.

**Figure 6 sensors-23-02249-f006:**
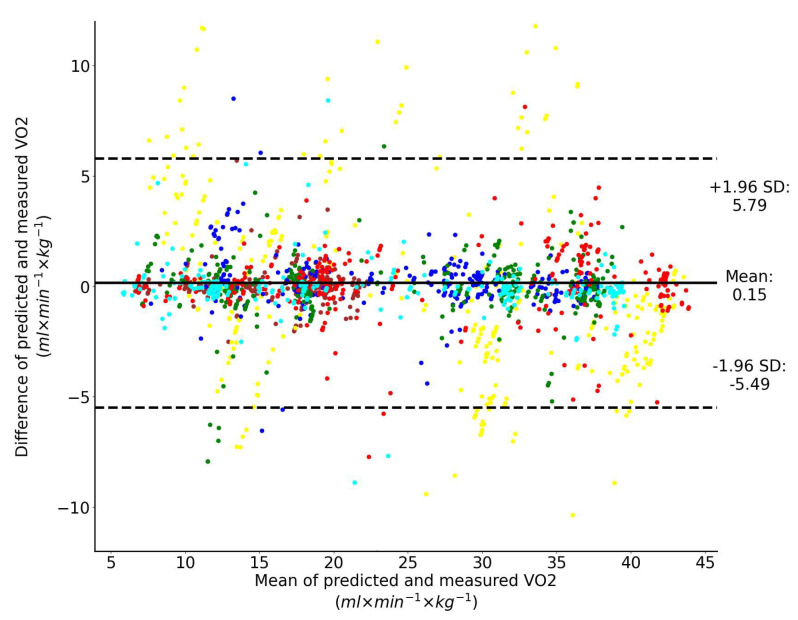
Bland–Altman analysis of the predicted and directly measured oxygen uptake with all exercise conditions combined. The inputs to the LSTM network include only speed. Dashed horizontal lines represent the 95% limits of agreement and the solid line represents the prediction bias. Each color represents data from a unique participant in the test set.

**Figure 7 sensors-23-02249-f007:**
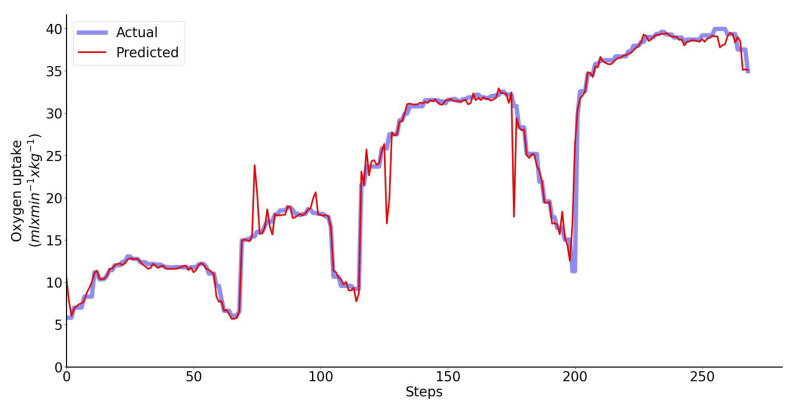
Performance of the regressor for a single representative athlete when only speed is used as the input to the LSTM network. Predicted (red) vs. measured (blue) oxygen uptake.

**Figure 8 sensors-23-02249-f008:**
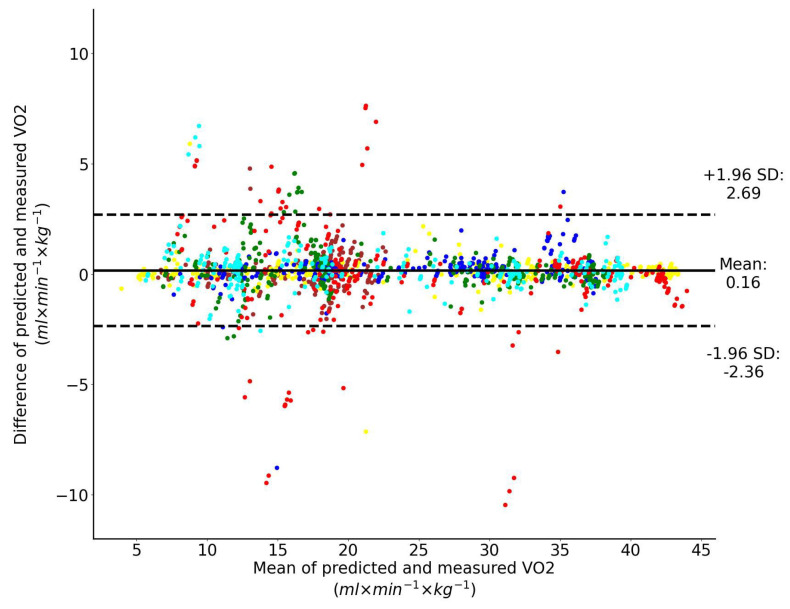
Bland–Altman analysis of the predicted and directly measured oxygen uptake with all exercise conditions combined. The inputs to the LSTM network include only heart rate. Dashed horizontal lines represent the 95% limits of agreement and the solid line represents the prediction bias. Each color represents data from a unique participant in the test set.

**Figure 9 sensors-23-02249-f009:**
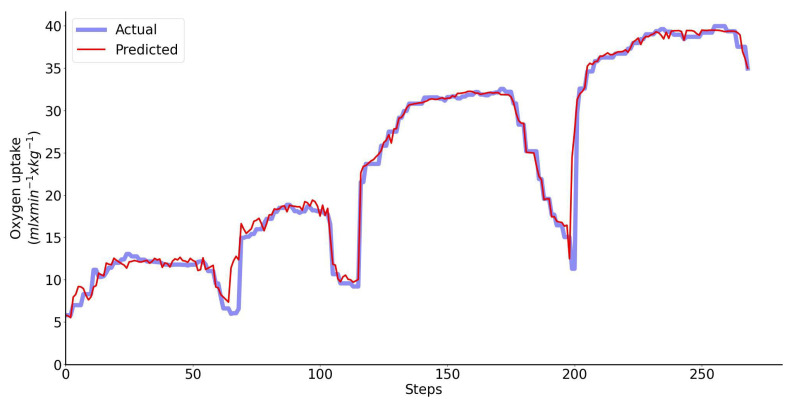
Performance of the regressor for a single representative athlete: predicted (red) vs. measured (blue) oxygen uptake. The inputs to the LSTM network include only HR.

**Figure 10 sensors-23-02249-f010:**
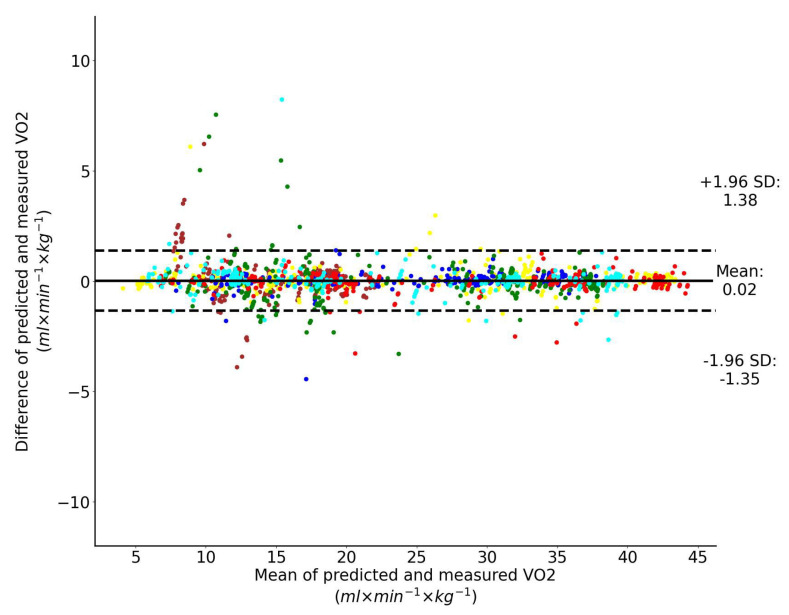
Bland–Altman analysis of the predicted and directly measured oxygen uptake with all exercise conditions combined. Dashed horizontal lines represent the 95% limits of agreement and the solid line represents the prediction bias. Each color represents data from a unique participant in the test set.

**Figure 11 sensors-23-02249-f011:**
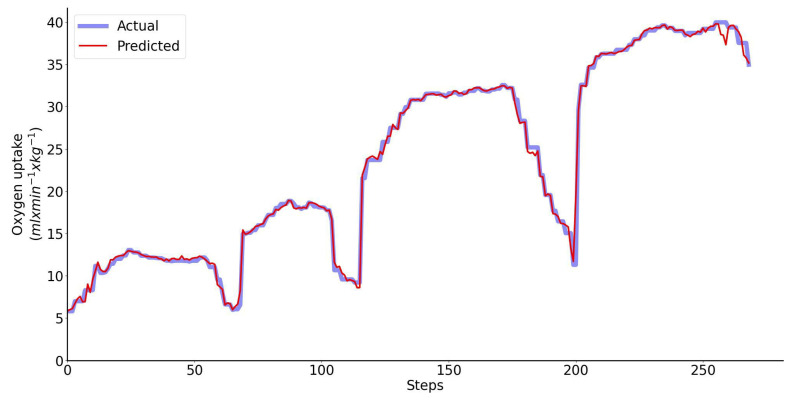
Performance of the regressor using a single representative athlete: predicted (red) vs. measured (blue) oxygen uptake. The inputs to the LSTM network include motion features and HR.

## Data Availability

The data presented in this study are available on request from the corresponding author.

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
