# Peer review of "Surrogate Modelling for Oxygen Uptake Prediction Using LSTM Neural Network"

_sensors, 2023, doi:10.3390/s23042249_

Round 1

Reviewer 1 Report

Manuscript ID: sensors-2161306

Title: Surrogate modelling for oxygen uptake prediction using LSTM neural network

Recommendation: Major revision

Brief summary

This manuscript presents an approach for the indirect estimation of oxygen uptake (VO2) through the exploitation of motion data, contributing to the estimation accuracy. The prediction of oxygen uptake is made through a long short-term memory neural network fed with data collected by means of a INS/GPS chest-worn device. Both walking and running are considered. The obtained accuracy suggests data combining motion and heart rate data can improve the VO2 estimation accuracy.

Broad comments

The topic is relevant, since oxygen uptake is an important physiological parameter not so straightforward to be measured in a continuous manner. However, no advancement in sensors is reported, since the authors focus on the application of pre-existent systems and exploit the measured data through AI-based analyses. Hence, the publication in Sensors journal should be carefully evaluate, maybe trying to better explain the work objective and the advancement beyond the state of the art.

The manuscript is written quite well and the English is generally fluent, even if some typos and errors are present. Hence, thoroughly re-reading the papers could help authors to correct them, as well as to optimize the punctuation use to improve readability.

The article should be better contextualized in the literature background; some references in the domain of Sensors could be added. Moreover, further discussion and comparison with the state of the art should be added in Section 4.

Some suggestions are provided in the next comments, which may help the authors in improving the quality of this paper.

Specific comments

Acronym for oxygen uptake: the “2” in “VO2” acronym should be placed as subscript. Please check all along the paper.

Abstract: the authors should quantify the value of the obtained accuracy and also provide an estimation for the improvement related to the combination of motion data to heart rate ones. Moreover, please always spell out acronyms when used for the first time (e.g. INS/GPS in the abstract).

Lines 15-19: please add proper bibliographic references.

Lines 47-50: please add some quantitative information on the accuracy of such predictions, indicating the related references.

Lines 51-56: please add an exemplificative reference on the use of LSTM network in this field.

Line 56: the authors should quantify the mentioned better performance of LSTM networks with respect to analytical models or linear regressions.

Lines 58-59: the authors should mention the exploited sensors and their accuracy., as well as report the performance of the tested network.

Lines 61-69: the performances obtained in the mentioned studies should be reported.

Lines 76-77: training a model based on a single person data could bias the model performance and also lead to scarcely generalizable models. Indeed, a robust model should be quite stable with respect to physiological variability. What can the authors state on this? A consideration should be added at least in the discussion section.

Lines 78-84: the authors should report the measurement accuracy related to the measured quantities, along with the performance of the tested predictive models.

Lines 95-96: the authors should report the demographics of the test population (at least age and BMI).

Lines 97-98: the WMA declaration of Helsinki should be referenced.

Line 100: on which criteria the heart rate monitor was or not applied in the trial? This should be clarified.

Lines 101-102: the measurement units of the International System of Units should be used. Please check.

Figure 1: labels related to the different sensors should be reported for ease of readability.

Lines 106-107: how the walking/running speeds were controlled on the track? This should be specified.

Lines 112-113: the authors should discuss the possible influence of these different conditions on the final results.

Line 114: the VO2 acronym has been already spelled out, hence the explanation is not needed here. The same for HR (line 120).

Lines 115-116: the spirometer accuracy should be reported. The same for the IMU+HR system (line 121).

Lines 119: the cut-off frequencies should be reported.

Line 123: the authors state that running and walking parameters were continuously measured; however, they should report the sampling frequency.

Figure 2: please check the caption, since the reported colours seem to not match the figure itself. Moreover, please avoid to report the time axis as “time*5”, since it could be misleading. Choosing seconds or minutes is better.

Lines 135-136: which is the accuracy in this type of synchronization? Why could the timestamps from the acquisition devices not be used to this aim? Please explain this aspect.

Lines 142-143: please report the sampling frequency and the method use for resampling.

Lines 147-152: the measurement units should be reported.

Line 221: the upper limit of the 95% confidence interval of the level of agreement is reported as “validity”, whereas the lower limit is no mentioned at all. The authors should revise this part.

Lines 228-229: the standard deviation reported here differs from what reported with Bland-Altman test. The authors should check and clarify this aspect.

Lines 230-236: is the Bland-Altman available for these considerations? The authors should add it.

Lines 232-233: please always report the measurement unit.

Line 250: please check the value of the upper limit of the confidence interval, since it does not coincide with that reported in the figure.

Lines 264-265: please quantify the improvement. The same for lines 287-288.

Lines 272-273: the obtained accuracy should be reported.

Lines 284: please quantify the obtained accuracy.

Lines 288-289: please report quantitative indications of this comparison.

Reviewer 2 Report

- This is an interesting paper on Surrogate modelling for oxygen uptake prediction using LSTM neural network. The write up needs some attention to its presentation with greater sophistication in the writing style and development of concepts with references and this could be a good paper but is missing careful construction as a paper trying to get across a message. Also how it fits into the existing literature is not clear to the reader so too much is left up to the reader to place it in the existing literature and you need to dig a bit harder into the literature and make sure you have a solid grasp on where your study fits, I do not see this clearly in the paper from start to finish, especially in the discussion.

 - The introduction is good and the readers can get an useful information to get an overview of the scientific question, but the discussion needs to better critique the studies used to support your approach and understanding of the question at hand for the reader. I find that the logic of the discussion could use some fine tuning as to setting up the picture you are trying to develop for the reader as to the problem you are trying to solve and how this study gives answers and insight into the practical applications for the reader.

- Make sure to include information about the sample, eleven recreational runners is not enough to clarify the context where you want to validate your method. A table with the sample information could facilitate the understanding of the different grouping variables of the sample.

- The information about the GPS is not provided in the first mention to the equipment used in the study. The measurement devices used must be described with their main characteristics.

- The discussion is not great and generally needs to be improved for clarity and preciseness. The references used are limited.

- First time the authors use acronyms, they must write the whole meaning.

- The references need to be reviewed according to the requirements of the journal.

- The tables need to be self-explanatory. the authors should avoid acronyms or abreviations.

Reviewer 3 Report

1.     Like the author said, heart rate is affected by factors like age, sex, fitness and environmental conditions. In the 2.1 part, I think it is better to add more information about runners, like age, weight, health condition.

2.     Can the author add more machine learning models to compare with the LSTM model proposed in this paper? Such as Random Forest, and SVM. I guess other models may also get good results.

Round 2

Reviewer 1 Report

Manuscript ID: sensors-2161306

Title: Surrogate modelling for oxygen uptake prediction using LSTM neural network

Recommendation: Minor revision

Overview

This is the revised version of a manuscript I previously revised. Most of my comments have been addressed and in my opinion the quality of the paper has been improved. Some further comments are provided below.

Specific comments

Line 94: please substitute “sec” with “s”. check this aspect all along the manuscript.

Model: regarding the fact that the model is based on data from a single subject, a consideration on this should be reported in the Discussion section, for the sake of completeness.

Line 120: please add the measurement unit for the BMI.

Walking/running speeds: the fact that they were chosen subjectively by the participants should be specified within the text.

Author Response

Dear reviewer,

We have addressed your comments and updated the final manuscript accordingly.

Reviewer 2 Report

I really appreciate the effort done for the authors, the review has been conducted in a proper way and for that reason I would like to accept the manuscript in present form

Author Response

The authors would also like to thank the reviewers of this paper for their detailed and thoughtful comments that considerably improved this work.

Reviewer 3 Report

Accept

Author Response

(The authors gave the same response as above.)
